# Extrahepatic Replication Sites of Hepatitis E Virus (HEV)

**Kush Kumar Yadav** [1,2,*] **and Scott P. Kenney** [1,2]

[1] Center for Food Animal Health, Department of Veterinary Preventive Medicine, The Ohio State University, Wooster, OH 43210, USA

[2] Center for Food Animal Health, Department of Animal Sciences, The Ohio State University, Wooster, OH 44691, USA

[*] Correspondence: yadav.94@osu.edu

**Simple Summary:** Hepatitis E virus is one of the emerging pathogens causing lethal effects to pregnant and immunosuppressed individuals. Recent progress in HEV demonstrated the ability of the virus to cross the natural body barriers such as blood–brain and blood–testis barriers. Extrahepatic existence of HEV was related to clinical manifestations via different case reports, case–control studies, and prospective studies. Knowledge about HEV-related extrahepatic diseases is very important for clinicians, as this would give them a clearer picture of organs involved in hepatitis E virus pathogenesis and spread. This summarization of the extrahepatic replication sites will help in designing treatment regimens and selection of samples for screening of hepatitis E viruses in cases of sporadic outbreaks.

**Abstract:** Hepatitis E virus (HEV) is an emerging viral disease known to cause acute viral hepatitis globally. Various genotypes of HEV have been identified that produce genotype specific lesions depending on the HEV targeted population. Pregnant or immunosuppressed individuals develop significantly more severe hepatitis E in comparison to the general population. In the last 40 years, we discovered that the tropism of HEV is not restricted to the liver, and virus replication was demonstrated in multiple organs. Out of the 10 body systems described in humans, HEV produces lesions causing a broad range of extrahepatic clinical manifestations in each of them. Affected body systems include nervous and musculoskeletal, cardiovascular, digestive, endocrine, integumentary, renal, respiratory, immune, and reproductive systems producing systemic lesions. All extrahepatic signs are caused by either direct HEV replication in these tissues, or indirectly by various immune mediated mechanisms. Extrahepatic replication features of HEV allowed it to cross the placental barrier, blood–brain barrier (BBB), and blood–testis barrier (BTB) that do not typically grant entry to viruses in general. Thus, in this review, we summarized the extrahepatic replication sites of HEV, listed the body systems where HEV invaded, and described multiple animal models including immunocompetent and immunosuppressed that were used to study the extrahepatic replication sites of HEV.

**Keywords:** hepatitis E; extrahepatic; clinical cases; animal models

## 1. Introduction

Hepatitis E virus (HEV) is the primary cause of acute viral hepatitis in humans [1]. Clinical manifestations include asymptomatic infection, generally seen in immunocompetent individuals, acute self-limiting hepatitis, persistent hepatitis in immunosuppressed and pregnant populations, and extrahepatic manifestations [2]. HEV was reported to cause about 70,000 deaths and 3000 stillbirths annually [3].

The recent reclassification of the family *Hepeviridae* comprises two subfamilies: *Orthohepevirinae* and *Parahepevirinae*. A collection of terrestrial and arboreal animals are included on the infection list of the four genera within the *Orthohepevirinae*. Most cases of hepatitis E

in humans are caused by the species *Paslahepevirus balayani*, which consist of eight geno-types (gt1–gt8), five of which are infectious to humans (gt1–gt4, gt7) [4,5]. HEV (gt1 and gt2) are obligated to humans, while HEV gt3 and HEV gt4 have zoonotic importance, as they travel via the food chain (pig and undercooked pork products) to develop a disease in humans [6]. *Avihepevirus*, *Rocahepevirus*, and *Chirohepevirus* are the other three genera predominantly circulating in birds, rodents, and bats, respectively. Notably, *Rocahepevirus* species *ratti* (rat HEV) was initially isolated from rats but was believed to have negligible zoonotic ability because of extreme genetic and antigenic divergence from HEV gt1 [7,8]. Recently, it was revealed that rat HEV can cause disease in humans via reports from Hong Kong [9–11].

HEV is a single stranded, positive-sense RNA virus with a 7.2 kb genome size. It is comprised of three open reading frames (ORFs), while some strains (*Paslahepevirus* gt1) also contain a fourth ORF, ORF4 [12–15]. The largest ORF in size is ORF1 (1693 amino acids) that encodes for nonstructural proteins (Burma strain). The translation of ORF2 (660 amino acids) and ORF3 (112–114 amino acids) is from the subgenomic RNA that encodes for structural proteins and a phosphoprotein/viroporin, respectively. Distinct functional domains: (a) methyl transferase (MT), (b) Y domain, (c) papain-like cysteine protease (PCP), (d) proline-rich hinge domain, (e) X domain, (f) RNA helicase, and (g) RNA-dependent RNA polymerase (RdRp) were reported in ORF1 based on computer-aided alignment and similarity prediction of the nucleotide sequence [2,16].

The shortage of appropriate in vitro models and in vivo models led to difficulties in understanding the pathogenesis of HEV failing to mimic the complete pathology demonstrated in humans [17,18]. Even though the fecal-oral route is considered as the main mode of HEV spread, the journey of virus particles from gastrointestinal tract to the liver and then to different organs has not been elaborated well. A very recent meta-assessment in 2018 revealed HEV prevalence up to 9% in the USA, 4.2% in Brazil, and up to 1% in the Caribbean [19]. Extrahepatic replication related to HEV acute and chronic infections have been reported by several publications [20]. Temporal associations between the infection and the extra-hepatic manifestations were made after eliminating other potential etiologies that could imitate these types of manifestations.

## 2. Extrahepatic Replication of HEV

### 2.1. Insights of Extrahepatic Replication

Even though HEV is known as a primary cause for acute hepatitis cases, chronic, and extrahepatic clinical diseases pertaining to different body systems cannot be neglected. Successful HEV replication in an organ can only be defined either by the presence of a negative-strand-specific reverse transcriptase PCR/in situ hybridization for replication complex RNA or via immunohistochemistry (IHC)/immunofluorescence assays (IFA) targeting the subgenomic RNA encoded proteins. Some HEV reports lacked the information on the negative sense RNA and IHC, limiting our complete understanding on the replication sites of HEV.

Multiple studies evaluating the vertical transmission of gt1 HEV in the fetus highlighted the successful in vitro HEV replication in the stromal cells [21], placental cells [22], both proven by IFA, and ex vivo replication in the maternal fetal interface recognized by in situ hybridization [23]. Additionally, HEV antigen was reported via IHC in the maternal and fetal side of the placental tissues collected from the HEV positive pregnant individuals after delivery [24].

The blood–testis barrier and blood–brain barriers limit immune cell trafficking into the immune privileged sites such as the testis and central nervous system, respectively [25]. There was very little knowledge about the replication of HEV in the immune privileged sites. Recently, the existence of gt3 HEV was demonstrated in the cerebrospinal fluid (CSF) [26] via the presence of a negative strand RNA in pigs and gt4 HEV via IHC in the macaque's testis [27].

The female reproductive organs' role in the HEV pathogenesis was one of the prime research interest areas for several years. Extensive research was conducted to understand the factors enhancing HEV virulence in pregnant women [2]. HEV tissue tropism in the ovary, ovum, and uterus was demonstrated in various species such as rabbits via IHC [28] and in BALB/c mice [29] via the detection of negative strand RNA. Even after 40 years of HEV discovery, the mechanisms behind HEV pregnancy mortality were not identified. Furthermore, pathology of HEV infection in the non-pregnant female reproductive system is completely unknown.

Of the various organs, the pancreas was extensively reported to harbor HEV replication. HEV was associated with 2.1% of acute pancreatitis cases, particularly in young males [30]. Experimental inoculation of miniature pigs with HEV gt3 demonstrated higher titers of HEV in the pancreas than the liver, highlighting necroptosis [31]. Interestingly, HEV antigens were described earlier in lymphoid tissues even before the noted classical organ of infection, "liver" via IHC [31].

It is very interesting to note that extrahepatic replication related to HEV is not limited to a genotype. From the listed studies, extrahepatic replication in males and non-pregnant females are related to gt3/gt4 HEV. One of the studies reported no evidence of gt1 HEV in the male reproductive system of humans [32]. However, an exception was seen when gt1 HEV acute infection was related to the digestive disorder, acalculous cholesystitis [33]. HEV gt1 was related with the female reproductive organs only during the pregnancy. Genotype specific lesions illustrate the need to understand the mechanisms behind the extrahepatic replication of HEV.

### 2.2. Pathogenesis of Extrahepatic Replication

There are many unknowns in the pathogenesis of extrahepatic manifestations due to HEV infection. Direct or indirect mechanisms were postulated with HEV induced pathogenesis. Direct mechanisms include HEV replication in the infected tissues, developing cellular damage. However, indirect mechanisms relate to cross-reactive immune triggers, development of immune complexes, or by indicated secondary infection [34]. Direct mechanisms were reported in vitro supporting the complete replication of HEV viral RNA and translation of viral capsid protein in some tissue types such as neuronal cells and human neuronal derived cells [35,36]. On the other side, humoral and cellular immune responses are responsible for the indirect mechanisms and are expected to be relevant in the pathogenesis of extrahepatic manifestations.

Remarkably, extrahepatic clinical manifestations are easily seen in immunocompetent patients rather than immunocompromised patients [37–39]. One report demonstrated that neurological manifestations were significantly more common in immunocompetent patients [n = 137] than immunocompromised patients [n = 63] (22.6% as compared to 3.2%) [37]. Similarly, a second report demonstrated neuralgic amyotrophy only in immunocompetent patients [38]. Likewise, pleiotropic neurologic disorders were reported in HEV-infected immunocompetent patients [39]. Such findings imply that immune-mediated mechanisms could be responsible for these extra hepatic disease manifestations.

Four decades of research advancements in HEV clearly demonstrated that the liver is not the only organ where HEV replication occurred. Neuronal cells, intestines, and human placenta are also important replication sites for HEV in humans [22,23,36,40,41]. Out of the 10 organ systems described in the human body, HEV was shown to affect all of these systems (nervous and musculoskeletal, cardiovascular, digestive, endocrine, integumentary, renal, respiratory, immune, reproductive systems) causing lesions. Thus, these extrahepatic manifestations need to be studied and characterized for the correct diagnosis of HEV infection. This review aims to summarize data pertaining to the extrahepatic manifestation of HEV reported in humans and recapitulated in animal models.

### 3. List of Body Systems Affected by HEV in Humans

Homeostasis is the ability of the body system to maintain a balance or equilibrium internally against external forces [42]. Invading organisms, such as HEV, trigger systemic physiological responses that disrupt normal homeostasis in the body systems when trying to establish itself in the host. Disturbance in homeostasis can be attributed to different clinical diseases which depend on the organ tropism of the pathogen and the subsequent pathogen–host–organ interactions.

Here, we list the system related clinical diseases that were demonstrated during HEV infection in humans.

### 3.1. Nervous and Musculoskeletal System

Neurologic manifestations of HEV are the most common extrahepatic manifestations and are increasingly being recognized as a complication of HEV infection. Europe (74%) and South East Asia Region (SEAR) nations (15%), mainly France, India, and Bangladesh reported multiple neurological disorders. Of the various disorders, the most common HEV neurological disorders are neuralgic amyotrophy (39%) and Guillain–Barre syndrome (37%) [43]. Numbers of patients reported with neurological disorders while having an HEV infection are presented below (Table 1).

**Table 1.** Nervous and musculoskeletal system disorders related to HEV infection.

| SN | Clinical Disease | Number of Patients | References |
|----|------------------|--------------------|------------|
| 1 | Neuralgic amyotrophy | 102 | [44–56] |
| 2 | Guillain–Barre syndrome | 36 | [39,57–65] |
| 3 | Myasthenia gravis | 1 | [66] |
| 4 | Polyneuromyopathy | 2 | [57,67] |
| 5 | Mononeuritis multiplex | 6 | [39] |
| 6 | Meningo-radiculitis | 5 | [39,68] |
| 7 | Cerebral ischemia | 5 | [46,65] |
| 8 | Epilepsy | 2 | [46] |
| 9 | Encephalitis | 5 | [46,48,65,69] |
| 10 | Facial Nerve Palsy | 3 | [46,70,71] |
| 11 | Encephalopathy | 1 | [72] |
| 12 | Encephalitic Parkinsonism | 1 | [73] |
| 13 | Transverse myelitis | 1 | [74] |
| 14 | Peripheral neuropathy | 3 | [57,65,75] |
| 15 | Vestibular neuritis | 1 | [57] |
| 16 | Myositis | 1 | [76] |

Of the various reported neurological disorders associated with HEV, myasthenia gravis, encephalopathy, encephalitic parkinsonism, vestibular neuritis were reported only in individual cases (Figure 1). Myasthenia gravis was reported in a 33-year-old immunocompetent female with acute HEV infection [66]. Encephalopathy was reported in a 66-year-old, immunosuppressed renal transplant woman [72]. Acute encephalitic parkinsonism was reported in a 17-year-old sportsman with acute HEV infection [73]. Vestibular neuritis was observed in a 92-year-old female with acute HEV infection.

Of the various musculoskeletal disorders related to HEV infection, transverse myelitis and myositis were observed only in single patients. Transverse myelitis was reported in a 62-year-old Caucasian woman with acute HEV infection [74]. Myositis was observed

in a 57-year-old man with a history of alcoholic chronic liver disease with acute HEV infection [76].

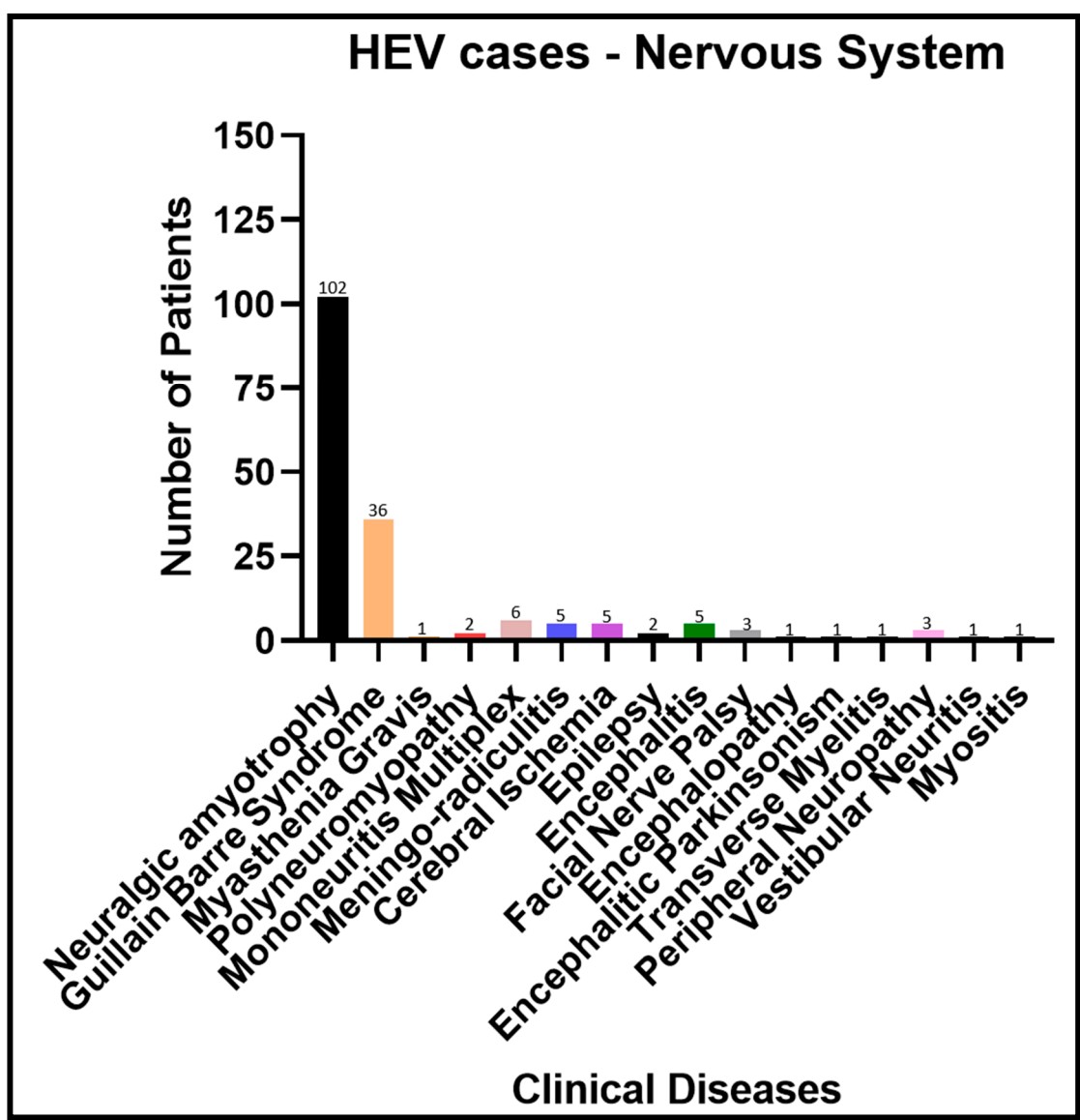

**Figure 1.** Summary of neurological and musculoskeletal clinical disorders presented in relation to HEV infection. The numbers are extracted from case reports, retrospective studies, case–control studies, case series, and cross-sectional studies. Every disorder and their references are listed in Table 1.

*3.2. Cardiovascular System*

Cryoglobulinemia is the most described cardiovascular disorder associated with HEV infection followed by monoclonal gammopathy, lymphocytosis, and thrombocytopenia (Table 2).

**Table 2.** Cardiovascular system disorders related to HEV infection.

| SN | Clinical Disease | Number of Patients | References |
|:--:|:--|:--:|:--:|
| 1 | Cardiac arrhythmia | 1 | [57] |
| 2 | Myocarditis | 2 | [77,78] |
| 3 | Anemia | 1 | [79] |
| 4 | Thrombocytopenia | 13 | [57,79] |
| 5 | Lymphocytosis | 14 | [57] |
| 6 | Lymphopenia | 8 | [57] |
| 7 | Leukocytosis | 1 | [30] |
| 8 | Massive hemolysis | 1 | [80] |
| 9 | Monoclonal gammopathy | 17 | [57] |
| 10 | Cryoglobulinemia | 51 | [81] |
| 11 | Long QT syndrome and Torsades de pointes | 1 | [82] |
| 12 | Metabolic acidosis | 1 | [30] |
| 13 | Acute myeloid leukemia | 2 | [57] |

Of the various cardiovascular system disorders, cardiac arrhythmia, anemia, leukocytosis, massive hemolysis, long QT syndrome, Torsades de Pointes, and metabolic acidosis were reported only in one patient (Figure 2). Cardiac arrhythmia was reported in a 73-year-old male with acute HEV infection [57]. Anemia was reported in a 5-year-old male with acute HEV infection. The child was coinfected with hepatitis B, and parvovirus [79]. Leukocytosis was reported in a 32-year-old male with acute HEV infection [30]. Massive hemolysis was reported in a 48-year-old male that resulted in renal failure with acute HEV infection [80]. Long QT syndrome and Torsades de Pointes was reported in a 62-year-old female with acute HEV infection [82]. Metabolic acidosis was reported in a 32-year-old male with acute HEV infection [30].

*3.3. Digestive System*

Pancreatitis is the most attributed digestive system disorder associated with HEV infection followed by pancreatic pseudocyst and acalculous cholecystitis (Table 3).

**Table 3.** Digestive system disorders related to HEV infection.

| SN | Clinical Disease | Number of Patients | References |
|:--:|:--|:--:|:--:|
| 1 | Pancreatitis | 22 | [30,83–86] |
| 2 | Pancreatic Pseudocyst | 1 | [87] |
| 3 | Acalculous cholecystitis | 1 | [33] |

Of the three digestive system disorders, pancreatic pseudocyst and acalculous cholecystitis associated with HEV were observed only in single patients (Figure 3). Pancreatic pseudocyst was observed in a 35-year-old man with acute HEV infection resulted in fatal outcome [87]. Acalculous cholecystitis was observed in a 24-year-old female with acute HEV infection [33].

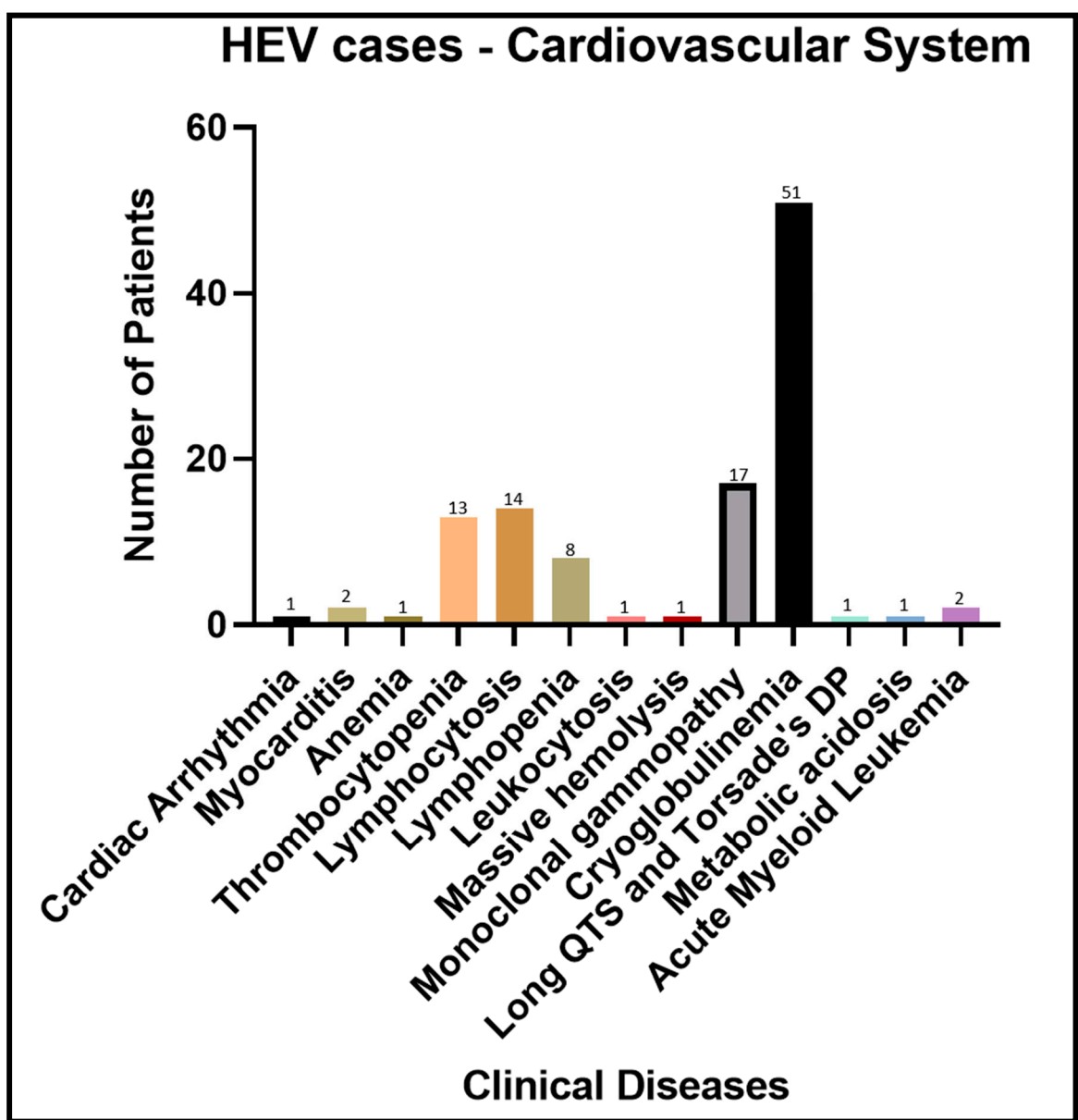

**Figure 2.** Summary of the cardiovascular clinical disorders presented in relation to HEV infection. The numbers are isolated from case reports, retrospective studies, case-control studies, case series, and cross-sectional studies. Every reported disorder and their references are listed in Table 2.

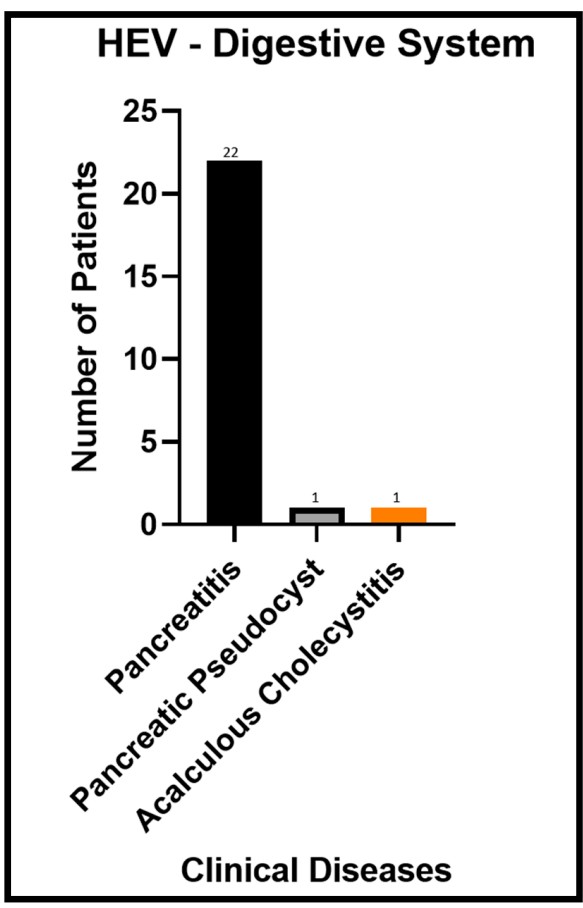

**Figure 3.** Summary of the digestive clinical disorders presented in relation to HEV infection. The numbers are isolated from case reports, retrospective studies, case-control studies, case series, and cross-sectional studies. Every disorder and their references were listed in Table 3.

### 3.4. Other Body Systems

An endocrine system disorder, autoimmune thyroiditis, was reported in a middle-aged woman with acute hepatitis E virus infection [88] (Table 4). An integumentary system disorder, cutaneous T cell lymphoproliferative, was reported in a 62-year-old Caucasian male with acute HEV infection [89]. A renal system disorder such as acute kidney injury was reported in a 32-year old Indian male with acute hepatitis infection [30]. Similarly, massive hemolysis causing renal failure was reported during acute hepatitis E infection in 48 year old man [80]. Cryoglobulinemic membranoproliferative glomerulonephritis, another renal system disorder, was reported in a 46-year-old man infected with gt3 HEV infection after the third kidney transplantation [90]. In contrast, another report of HEV induced cryoglobulinemic glomerulonephritis was from non-immunocompromised 48-year-old native French man [91].

Interestingly, a respiratory system disorder, pleural effusion was reported during coinfection of hepatitis E and hepatitis A viruses [92]. Immune system disorders, acute allograft dysfunction was reported in a 56-year-old male patient after 17 years of liver transplant with acute HEV infection [93]. Histologic reports of his liver demonstrated portal infiltration by lymphocytes during the acute HEV infection [93]. Reproductive system disorder, infertility was reported in 185 males of which 28.11% (52/185) were HEV positive [27]. Also, it was found that in some patients the blood–testis barrier was destroyed by HEV infection [27].

**Table 4.** Other body systems disorders related to HEV infection.

| SN | Clinical Disease | Body System | Number of Patients | References |
|----|------------------|-------------|--------------------|------------|
| 1 | Autoimmune Thyroiditis | Endocrine | 1 | [88] |
| 2 | Cutaneous T-Cell lymphoproliferative disorder | Integumentary | 1 | [89] |
| 3 | Acute Kidney Injury | Renal | 2 | [30,80] |
| 4 | Cryoglobulinemic membranoproliferative glomerulonephritis | Renal | 2 | [90,91] |
| 5 | Pleural Effusion | Respiratory | 1 | [92] |
| 6 | Acute Graft Dysfunction | Immune | 1 | [93] |
| 7 | Infertility | Reproductive | 52 | [27] |

## 4. List of the Animal Models Studied with Involved Body Systems and Extrahepatic Replication Sites for HEV

To dissect the in vivo characteristics of HEV and determine its replicative sites other than the liver, it is important to understand the viral pathogenesis utilizing various animal models. Multiple animal models were used to understand the molecular pathobiology of HEV and to gain an understanding of the extrahepatic replication sites' role in invading the immune system of the host. Although immunocompetent animal models depict the susceptibility to distinct HEV strains and mimic human disease [94], recapitulating the immunosuppressed individual's scenario can only be done utilizing the immunosuppressed animal models. Thus, immunosuppressed animal models gained popularity mimicking the immunosuppressed solid organ transplant recipients. Such immunosuppressed animal models can be used to understand the disease severity and the role of extrahepatic sites in the maintenance of HEV. Below, we list the HEV genotypes and/or strains, body systems, and extrahepatic sites involved during the infection in immunocompetent and immunosuppressed animal models (Table 5).

**Table 5.** List of animal models used to study HEV extrahepatic replication.

| SN | Animal Model | HEV Genotype/Strain | Body System Involved | Extrahepatic Sites | Demonstration of Viral Replication Via | References |
|----|--------------|---------------------|----------------------|--------------------|-----------------------------------------|-----------|
| 1 | Mongolian Gerbil | Human Gt4 | Digestive, Renal, Lymphatic | Kidneys, Spleen, Small Intestine | IHC and positive strand RT-qPCR (not confirmatory) | [95] |
| | | Gt1 (clinical human sample) | Digestive, Renal, Lymphatic | Spleen, Kidney | Positive strand RT-qPCR (not confirmatory) | [96] |
| | | Human Gt3 | Digestive, Lymphatic | Spleen | IHC and positive strand RT-qPCR (not confirmatory) | [97] |
| | | Swine HEV Gt4 (CHN-HB-HD-L2) | Digestive, Renal, Lymphatic, Respiratory, Reproductive | Brain, Spinal cord, Spleen, Peripheral blood monocytes, Pancreas, Lung, Lymph node, Kidney, Duodenum, Jejunum, Ileum, Colon, Placenta, Urine | IHC and negative strand RT-qPCR | [98] |

**Table 5.** *Cont.*

| SN | Animal Model | HEV Genotype/Strain | Body System Involved | Extrahepatic Sites | Demonstration of Viral Replication Via | References |
|---|---|---|---|---|---|---|
| 2 | Pregnant Rhesus Macaques | Gt4 (KM01) | Digestive, Fetal (Digestive, Renal) | Spleen, Kidneys, Intestine | IHC and positive strand RT-qPCR (not confirmatory) | [99] |
| | Cynomolgus Macaques | Human derived HEV | Digestive | Not listed | Positive strand RT-qPCR (not confirmatory) | [100] |
| | | Human Gt3 | Digestive, Nervous, Lymphatics | Bone marrow | IHC and negative strand RT-qPCR | [101] |
| | Immunocompromised Cynomolgus Monkey | Gt3 | Digestive, Lymphatic | Spleen, Duodenum, Colon, Lymph node, Pancreas | IHC and negative strand RT-qPCR | [102] |
| 3 | Pig | unknown | Digestive | Not listed | Visualization of virus-like particles | [103] |
| | | Swine HEV | Digestive, Respiratory | Not listed | Positive strand RT-qPCR (not confirmatory) | [104] |
| | | US2 | Digestive | Not listed | Positive strand RT-qPCR (not confirmatory) | [105] |
| | Immunosuppressed pigs | Human HEV (US2 strain), Gt3 | Digestive, Immune | Not listed | Positive strand RT-qPCR (not confirmatory) | [106] |
| 4 | Miniature pigs | Gt3 | Digestive, Endocrine, Respiratory, Renal, Nervous | Pancreas, Kidney, Brain, Peyer's patches, Lungs (bronchioles) | IHC and positive strand RT-qPCR (not confirmatory) | [31,107] |
| 5 | Rabbit | Swine derived HEV Gt4 | Digestive, Lymphatic, | Spleen | Negative strand RT-qPCR | [108] |
| | | Rabbit derived HEV Gt3 | Digestive, lymphatic | Spleen | Negative strand RT-qPCR | [108] |
| | SPF rabbits | Rabbit HEV | Digestive, Renal, Lymphatic, Respiratory, Nervous | Stomach, Duodenum, Kidney, Bile, Lung, Bladder, Brain | IHC and negative strand RT-qPCR | [109] |
| | | Swine Gt4 | Mild digestive | Not listed | IHC and negative strand RT-qPCR | [109] |
| | Pregnant rabbits | Rabbit HEV (CHN-BJ-R14) | Digestive, Reproductive | Placenta | IHC and negative strand RT-qPCR | [110] |
| | | Rabbit gt3 (KOR-Rb-1) | Digestive, Reproductive | Uterus | Positive strand RT-qPCR (not confirmatory) | [28] |
| | Immunosuppressed Rabbit | Rabbit derived HEV-3ra | Digestive, Renal, Nervous | Kidney, Duodenum, Jejunum, Cecum, Colon, Urine, Cerebrospinal fluid | Positive strand RT-qPCR (not confirmatory) | [111] |
| 6 | BALB/c nude mice | Swine HEV Gt4 | Digestive, Lymphatic, Renal | Spleen, Kidney, Jejunum, Ileum, Colon | IHC and positive strand RT-qPCR (not confirmatory) | [112] |
| | Pregnant BALB/c mice | Swine derived Gt4 (KM01) | Digestive, Reproductive, Renal, Placenta and neonatal liver | Spleen, Kidney, Colon, Uterus, Placenta | IHC and negative strand RT-qPCR | [113] |
| 7 | Rat | Human feces derived (TK-037/92) | Digestive, Lymphatic | Spleen, Mesenteric lymph nodes, Small intestine | IHC and positive strand RT-qPCR (not confirmatory) | [114] |
| | Immunocompromised rats | Human derived rat strain (CCY) | Digestive | Not listed | IHC and positive strand RT-qPCR (not confirmatory) | [115] |

### 5. Notable Abilities of HEV

In general, reorganization of intracellular membranes (endoplasmic reticulum, golgi apparatus, mitochondria, endosomes and/or lysosomes) to establish replication sites is the unique attribute of positive-stranded RNA viruses. These replication complexes function as platforms to enhance viral and cellular cofactors at local level and offer a secure setting which decreases the innate immune system detection of viral proteins and nucleic acids [116]. Genetic plasticity is one of the advantageous attributes of RNA viruses, thus, they can promptly produce drug-resistant viral populations. This helps them to escape the host immune system under the pressure of homeostasis. Lack of proofreading attributed to RdRp is responsible for such variability [117]. A report suggested that during viral replication, hundreds of progenies (quasispecies) are produced with a mutation rate comprising $10^{-6}$ to $10^{-4}$ substitutions per nucleotide resulting in the alteration of genomic sequences attributed to one or few nucleotide substitutions [118]. In addition, the propagation of a well-adapted viral population depends upon the quasispecies fitness ability indicating Darwinian evolution and natural selection [118].

ORF1 encodes the replicative machinery of HEV which contains methyltransferase, RNA helicase, X and Y domains, protease, and RdRp [119]. Polyprotein processing into distinct domains during the HEV life cycle remains debatable but is generally seen in other viruses [120–126]. Moreover, the hypervariable region (HVR) of the HEV ORF1 exhibits sequence differences even among isolates of identical virus genotypes [127]. Acute hepatitis patients were reported to consist of HEV with naïve genomic rearrangements associated with HEV persistence and long term use of ribavirin. It resulted in ribavirin insensitivity to HEV with an observed increase in the HEV quasispecies [128–132]. A recombination was demonstrated between HEV genotypes and fragments of human genes and HEV strains. For example, human RPS17 (ribosomal protein S17) inclusion in the hypervariable region boosted replication in hepatoma cells [133].

In addition, HEV is found in two forms: nonenveloped and quasi-enveloped. Both were demonstrated to take different replicative cycles with differences in infectivity [134]. For example, effective replication of KM01 (genotype 4 HEV) strain isolated from pig fecal samples was demonstrated in A549 cells and Huh7.5.1 cells following an extensive period (more than 11 years) of serial passaging. Feces derived or bile-derived HEV (nonenveloped) and cell culture-derived (enveloped) HEV was reported in A549 cells when inoculated with the KM01 strain. Fairly distinct infectivity was noted between two forms (5.34% of enveloped HEV versus 10.80% of non-enveloped HEV in A549 cells) suggesting non-enveloped HEV is more infectious when compared to enveloped HEV [134]. Unlike non-enveloped HEV, the entry of enveloped HEV needs Rab5 and Rab7, small GTPases participating in endosomal trafficking [135]. Interestingly, the infectivity of enveloped HEV was enhanced when the virus was treated with 1% NP-40 to eliminate the quasi-enveloped membranes [134].

The in vitro blood-brain barrier (BBB) model and in vivo HEV infection study in pigs revealed HEV virions, both quasi-enveloped and nonenveloped, possessed the ability to compromise the BBB and enter the central nervous system (CNS) [26]. Furthermore, HEV can cross the placental barrier and was demonstrated to transmit via vertical transmission [23,24]. In addition, HEV is shown to be persistent in the ejaculate of chronically infected men [27,136]. No noted difference could be observed between enveloped HEV, when compared morphologically to serum derived HEV particles, suggesting its ability to cross the blood–testis barrier (BTB) [26,27,136]. Interestingly, HEV was also reported in the muscle of wild boars [137–139]. Thus, with the increase in host organ tropism, HEV is slowly becoming an even more important emerging pathogen. HEV currently ranks 6th on a list of 50 priority zoonotic viral pathogens, only behind Lassa, SARS-CoV-2, Ebola, Seoul, and Nipah viruses, and ahead of viral pathogens such as Monkeypox, Marburg, Rabies, Simian immunodeficiency virus, and other coronaviruses currently of global relevance [140]. HEV being entrenched as a food borne zoonosis possesses a higher spillover risk estimate

than some known zoonotic pathogens which necessitates a better understanding of HEV pathogenesis and its virulence mechanisms in relation to body system preference.

## 6. Conclusions

The increasing host range of HEV guided numerous innovative experimental and naturally existing animal models capable of recapitulating extrahepatic manifestations. These models offer prospects for potential HEV study with strains from humans, macaques, pigs, wild boar, rabbits, gerbils, and mice that could be used to understand the extrahepatic replication role in HEV pathogenesis. In addition, there could be multiple reservoir cells for HEV in the human body that still need to be investigated for the application of proper treatment and prevention strategies against HEV. Summarizing the body systems involved in the HEV replication gives us a somewhat clearer picture of organs involved in HEV pathogenesis and spread. This summary will help in designing treatment regimens and selection of samples for screening of HEV in cases of a sporadic outbreak.

**Author Contributions:** Conceptualization, K.K.Y. and S.P.K.; writing-original draft preparation, K.K.Y.; writing—review and editing, K.K.Y. and S.P.K. All authors have read and agreed to the published version of the manuscript.

**Funding:** Additional salaries and research support were provided by state and federal funds appropriated to the Ohio Agricultural Research and Development Center, The Ohio State University, and from the research funds of National Institute of Allergy and Infectious Diseases (#R21AI151736).

**Institutional Review Board Statement:** Not applicable.

**Informed Consent Statement:** Not applicable.

**Data Availability Statement:** Not applicable.

**Conflicts of Interest:** The authors declare no conflict of interest.

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
